# Dietary Exposure of the Taiwan Population to Mercury Content in Various Seafood Assessed by a Total Diet Study

**DOI:** 10.3390/ijerph182212227

**Published:** 2021-11-21

**Authors:** Pinpin Lin, Fan-Hua Nan, Min-Pei Ling

**Affiliations:** 1National Institute of Environmental Health Sciences, National Health Research Institutes, Zhunan Town, Miaoli County 35053, Taiwan; pplin@nhri.org.tw; 2Department of Aquaculture, National Taiwan Ocean University, Keelung City 20224, Taiwan; fhnan@mail.ntou.edu.tw; 3Department of Food Science, National Taiwan Ocean University, Keelung City 20224, Taiwan

**Keywords:** methylmercury, pelagic fish, farmed fish, health risk assessment, total diet study

## Abstract

This paper examines the health risks of exposure to methylmercury (MeHg) through the consumption of mercury-contaminated seafood in Taiwan, based on the total diet study (TDS) method. Samples of seafood (*n* = 140) were purchased at fishing harbors or supermarkets and classified into seven categories (pelagic fish, inshore fish, farmed fish, shellfish, cephalopods, crustaceans, and algae). For each sample, we analyzed raw and cooked versions and compared the concentration difference. Total mercury (THg) was detected at the highest rate and in the highest concentrations in pelagic fish, followed by inshore fish and other farmed fish. The average concentration of THg was higher after cooking. In a 75th percentile scenario, the hazard indices for children aged 1 to 3 years and children aged 4 to 6 years were higher than 100% of the provisional tolerable weekly intake. Taking into consideration the risk assessment results, MeHg concentrations, and the nutritional composition of fish, we have provided weekly consumption advisories for children aged 1 to 3 years, children aged 4 to 6 years, and childbearing women aged 19 to 49 years. The weekly consumption advisories for childbearing women are 35 g/week of pelagic fish and 245 g/week of inshore fish based on the risk results from MeHg and the potential benefits from eicosapentaenoic acid (EPA) and docosahexaenoic acid (DHA) intake.

## 1. Introduction

Mercury (Hg) is a metal that is released into the environment from both natural and anthropogenic sources. Human activities are the main source of Hg released into the environment, particularly coal-fired power stations, residential coal burning, industrial processes, waste incinerators, and as a result of mining for Hg, gold, and other metals. Once released, Hg undergoes a series of complex transformations, and Hg cycles between the atmosphere, oceans, and land. Eventually, most of these contaminants enter water bodies, then into aquatic organisms through bioaccumulation, and build up at higher concentrations in animals near the top of the food chain. The three chemical forms of total mercury (THg) are elemental or metallic Hg, inorganic Hg, and organic Hg. Methylmercury (MeHg) is by far the most common form of organic Hg in the food chain. The MeHg concentration is highest in irrigation canal sediments; rivers are second, followed by wetlands [1]. MeHg is of particular concern as MeHg can build up in certain edible freshwater and saltwater fish and marine mammals to levels that are many times greater than levels in the surrounding water [2,3].

MeHg is extremely harmful to humans. Excessive MeHg intake damages the human nervous system [1,4,5,6]. In particular, MeHg consumed by pregnant women negatively affects the growth of the fetus and delivery [5,7]; in children, MeHg intake may lead to impaired development, including development of the nervous system and compromised cognition [1,8,9]. Maternal fish intake was determined to be a possible cofactor during an investigation of the association between prenatal MeHg exposure and child development [10].

There have been a great number of environmental studies concerning Hg contamination in Taiwan and in countries overseas, with many results showing that high concentrations of Hg can accumulate in aquatic organisms [11,12,13,14]. Taiwan is surrounded by seas that offer a rich range of seafood. Taiwanese people are able to easily acquire seafood as a source of dietary protein [15]. While this seafood contains nutrients of excellent quality, such as proteins, ω-3 polyunsaturated fatty acids (ω-3 PUFAs), vitamins, and minerals [4,5,8,16], excessive consumption may result in excessive MeHg intake, which can cause potential health safety risks. Therefore, we conducted a large-scale total diet study (TDS) on seafood.

TDSs have been used to measure the dietary intake of specific analytes by population groups in countries or defined regions and to assess associated public health risks due to chronic exposure. The TDS, also known as a “market basket study”, has been used as a national monitoring research tool for food contamination and dietary exposure. It is based on a national representative of food consumption for different subgroups of the general population. TDS representative foods are designed to measure the average intake of chemicals found in cooked or processed foods. In a TDS, food samples are prepared prior to analysis as they would be consumed (table-ready) so that the analytical results provide the basis for a realistic estimation of the dietary intake of these analytes and provide a reasonable assessment of the health risks [17,18].

Many studies [19,20,21] have indicated large amounts of THg or MeHg accumulate in aquatic organisms. The European Food Safety Authority (EFSA) conducted a study in 2012 analyzing the THg levels present in 20 different food categories [22], including grains and grain-based products, vegetables and vegetable products, starchy roots and tubers, legumes, nuts and oilseeds, fruit and fruit products, meat and meat products, fish and other seafood, milk and dairy products, eggs and egg products, sugar and confectioneries, animal and vegetable fats and oils, fruit and vegetable juices, non-alcoholic beverages, alcoholic beverages, drinking water, herbs, spices and condiments, foods for infants and small children, products for special nutritional use, composite foods, snacks, desserts, and other foods. According to the average THg levels measured in that study, fish and other seafood had an average THg concentration of 133 µg/kg, which was the highest among all 20 food categories. Most of the remaining food categories were below 5 µg/kg. The three food categories with the lowest THg levels were drinking water (0.1 µg/kg), starchy roots and tubers (0.8 µg/kg), and eggs and egg products (1.2 µg/kg).

Humans are exposed to MeHg through fish intake as Hg can transform into MeHg in aquatic environments. According to recent studies, fish intake is considered a major pathway for exposure to MeHg [23,24,25]. MeHg has a high affinity for sulfhydryl protein groups. When humans ingest MeHg, it interacts with glutathione to form a MeHg-glutathione compound, which is distributed to various tissues and organs through the blood [4].

Taiwan has rich and abundant aquatic resources as it is surrounded by the ocean. Most fish, shellfish, cephalopods, and crustaceans are at the top of aquatic food chains [26]. In addition to the accumulation of MeHg through the water and sediment, the accumulation of hazardous metals may occur through the biomagnification effect of the food chain. Hazardous metals accumulate readily in sediment; benthic shellfish, crustaceans, and cephalopods, which are in contact with the sediment for long periods, subsequently accumulate hazardous metals. In order to ensure food safety for the public, different types of fresh seafood should be tested, and a database on background concentrations of MeHg in seafood should be established in order to understand the current exposure of the public [27,28].

The purposes of this study were: (1) to develop a representative list of seafood using the TDS method and analyze the THg levels in different seafood purchased from various representative fishing ports of Taiwan, (2) to estimate the human health risk from MeHg in different types of cooked seafood, and (3) to evaluate the consumption advisories on large-sized fish for the Taiwanese population based on the MeHg risk assessment results.

## 2. Materials and Methods

### 2.1. Constructing a Representative Diet List

Classification principles for seafood: On the basis of the Sanitation Standards for Aquatic Animals and the Sanitation Standards for Algae Foods issued by the Ministry of Health and Welfare in Taiwan, we divided aquatic organisms into “pelagic fish”, “inshore fish”, “other farmed fish”, “shellfish”, “cephalopods”, “crustaceans”, and “algae”. These seven categories can be subdivided into 82 seafood species, which were represented by 140 samples (Table 1).

Representative list of seafood: The establishment of the representative list of seafood was mainly based on four principles. The first principle was that the aquatic animal category in the Sanitation Standards for Aquatic Animals had to be a major item. The second principle was that on the basis of the Sanitation Standards for Algae Foods, the algae had to be a major item. The third principle was the domestic sales volume, based on domestic production and imports minus exports in the 2015 Fisheries Industry Statistics. Products were ranked from high to low by domestic sales volume (tons). The fourth principle was that on the basis of the Nutrition and Health Survey in Taiwan (NASHIT), only seafood with relatively high consumption rates were included in our list.

### 2.2. Planning the Sampling Strategy

Selection of sampling sites: First, the country’s administrative districts were divided into four regions (northern, western, southern, and eastern), which covered 16 cities and counties. The chosen locations were divided according to the products purchased. Seafood was divided into imported and non-imported products. Non-imported seafood was purchased in the cities and counties with the three highest rankings for sales volume according to the 2015 Fisheries Industry Statistics. For imported seafood and processed seafood, we took the city or county with the highest population in northern, western, southern, and eastern Taiwan, respectively. The counties and cities where this study purchased samples were: (1) in the northern region: Keelung City, Taipei City, New Taipei City, Taoyuan County, and Hsinchu County; (2) in the western region: Miaoli County, Taichung City, Changhua County, Yunlin County, and Chiayi County; (3) in the southern region: Tainan City, Kaohsiung City, and Pingtung County; and (4) in the eastern region: Yilan County, Taitung County, and Hualien County.

The sampling sites were selected based on places where people regularly shop, major fishing ports in each of the cities and counties, traditional wet markets or afternoon markets, and supermarkets or discount stores. We selected well-known places with relatively high numbers of consumers.

### 2.3. Analysis of THg Concentrations in Seafood

For the samples of seafood, the scales, viscera, skin, and bones of the fish were removed, and the edible parts were homogenized. All stainless-steel laboratory equipment was soaked the night before in a 32% nitric acid solution to remove impurities and metal ions. The 140 seafood samples were each separated into two testing conditions, raw or steam-cooked, to compare the differences in THg concentration before and after cooking. For steam-cooking, samples were placed in a pot and steamed for 10 min. The THg concentration was analyzed with inductively coupled plasma mass spectrometry (ICP-MS). The mean recovery of THg in the certified reference material was 94.85%. The detection limit for THg analysis was 0.02 mg/kg.

### 2.4. Seafood Safety Risk Assessment

The hazard quotient (HQ) values in this study were the ratio between the estimated daily intake (EDI) and the provisional tolerable weekly intake (PTWI), expressed as 100%PTWI. HQs are used to assess the non-carcinogenic human health risks associated with MeHg in seafood, as described in the following equation:(1)$$\mathrm{HQ}\;(\%\mathrm{PTWI})=\frac{\mathrm{EDI}}{\mathrm{PTWI}}=\frac{C\times \mathrm{CR}\times F_{c}}{\mathrm{BW}\times \mathrm{PTWI}}\times 100\%$$
where C is the MeHg concentration in the samples of the seven categories of seafood (mg/kg), CR represents the consumption rate of the seven categories of seafood among the different age groups (g/day) (Table 2, F_c_ represents the conversion factor of period which is equal to 7 (week/day), and BW is body weight of the different age groups (kg). PTWI is the MeHg reference dose of 1.6 µg/kg BW/week [29]. The daily seafood CR of all exposure groups was based on values taken from the Nutrition and Health Survey in Taiwan (NAHSIT). The THg concentration in steam-cooked seafood was used to calculate the MeHg concentration based on the ratio of MeHg in THg (%MeHg/THg) which were referred to in the previous studies [30,31,32,33,34,35,36,37,38] collated in Table 3.

The hazard index (HI) for MeHg in seafood was calculated by the sum of the individual HQ values for pelagic fish, inshore fish, other fish, shellfish, cephalopods, crustaceans, and algae.
(2)HI=∑HQ
where HI is the total of the HQs of the seven classifications of seafood among the different age groups.

### 2.5. Risk-Based Consumption Advisories

We proposed the consumption advice for MeHg based on estimates of mean HQ risks and on an assumption of the acceptable risk being equal to 100%PTWI. This study calculated the weekly consumption rates and resulting MeHg exposure of various population groups, including children aged 1 to 3 years and 4 to 6 years, and childbearing women aged 19 to 49 years. Swordfish and tuna were characterized by a higher MeHg concentration and lower eicosapentaenoic acid (EPA) and docosahexaenoic acid (DHA) levels, whereas salmon, mackerel, and greater amberjack had lower levels of MeHg and higher levels of EPA and DHA [5,39,40]. Here, we set up hypothetical scenarios where pelagic fish were not consumed or were consumed at one (35 g per serving) or two servings per week, and inshore fish were consumed at 0.5 or one serving (35 g per serving) and provide recommendations for weekly consumption advisories for other farmed fish for different exposed population groups.

### 2.6. Uncertainty Analyses

In exposure and risk assessment, there are several sources of uncertainty. Due to inherent natural variability, model variables can be defined in terms of a probability density function that is derived from a limited set of observations. The data, however, may not be representative of the entire population, and sample statistics may not be accurate estimates of the true values of the population parameters, i.e., C, CR, and BW. This leads to uncertainty in the parameter estimation procedures. To explicitly account for this uncertainty/variability and its impact on the estimation of expected risk, a Monte Carlo (MC) simulation was adopted. To test the convergence and the stability of the numerical output, we performed independent runs at 1000, 4000, 5000, and 10,000 iterations, with each parameter sampled independently from the appropriate distribution at the start of each replicate. Largely due to limitations in the data used to derive model parameters, inputs were assumed to be independent. The result showed that 10,000 iterations were sufficient to ensure the stability of results. The MC simulation and sensitivity analysis were implemented using Crystal Ball^®^ (Version 11.1; Oracle Corporation, Redwood Shores, CO, USA). We incorporated probability distributions into the MC simulation to obtain 5–95th percentiles for all uncertainty analyses.

## 3. Results and Discussion

### 3.1. THg and MeHg Concentration in Seafood

On the basis of the list of representative seafood, we divided the seafood into seven categories subdivided into 82 species, including 140 raw and 140 cooked (steamed) samples (Table 4). To conservatively estimate the concentrations of THg and MeHg, according to the assumptions applied by the World Health Organization (WHO) European Programme for Monitoring and Assessment of Dietary Exposure to Potentially Hazardous Substances (GEMS/Food-EURO), if the proportion of analytical results that are non-detectable (ND) is less than 60%, the ND results should be replaced with LOD/2 [41]. For this study, the proportion of ND analytical results was less than 60%, so we calculated the average concentration by replacing the ND results with LOD/2. In addition, we also investigated the ratio of MeHg to THg in different seafood in the literature to estimate the MeHg concentrations in the different seafood used in the subsequent risk assessment calculations. In the literature, nine articles related to the conversion rate of THg to MeHg and 27 families of seafood were previously sampled [30,31,32,33,34,35,36,37,38]. If the literature provided a conversion rate for a particular family, then that 100%MeHg/THg was used. If there was no recommendation in the literature, then we used the recommendations from the EFSA report [22]: a 100%MeHg/THg of 100% for fish and 80% for crustaceans. The 100%MeHg/THg used here were 73–100% for pelagic fish, 83.6–100% for inshore fish, 75–100% for other farmed fish, 35–82% for shellfish, 72.8–92% for cephalopods, and 80% for crustaceans and algae, respectively (Table 3). The THg and MeHg concentrations in the seven seafood categories are described below.

The concentrations of THg in raw samples ranged from 0.03–3.16 mg/kg in pelagic fish; not detected (ND)–0.78 mg/kg in inshore fish; ND–0.35 mg/kg in other farmed fish; ND–0.06 mg/kg in shellfish; ND–0.05 mg/kg in cephalopods; ND–0.14 mg/kg in crustaceans; and ND in algae. The MeHg concentrations in raw samples ranged from 0.025–2.307 mg/kg in pelagic fish; ND–0.760 mg/kg in inshore fish; ND–0.329 mg/kg in other farmed fish; ND–0.048 mg/kg in shellfish; ND–0.036 mg/kg in cephalopods; ND–0.112 mg/kg in crustaceans; and ND–0.024 mg/kg in algae. The concentrations of THg were higher in pelagic fish, followed by inshore fish. The detection rates in different categories were pelagic fish (100%), inshore fish (91%), other farmed fish (62%), shellfish (33%), cephalopods (50%), crustaceans (69%), and algae (17%). Pelagic fish and inshore fish had higher THg detection rates, while algae had less, but all the seafood categories were detected THg in this study.

In cooked samples, the concentrations of THg ranged from 0.05–4.59 mg/kg in pelagic fish; ND–1.39 mg/kg in inshore fish; ND–0.47 mg/kg in other farmed fish; ND–0.05 mg/kg in shellfish; 0.02–0.11 mg/kg in cephalopods; ND–0.23 mg/kg in crustaceans; and ND–0.01 mg/kg in algae. The MeHg concentrations in cooked samples ranged from 0.042–3.351 mg/kg in pelagic fish; ND–1.354 mg/kg in inshore fish; ND–0.442 mg/kg in other farmed fish; ND–0.0400 mg/kg in shellfish; 0.016–0.080 mg/kg in cephalopods; ND–0.184 mg/kg in crustaceans; and ND–0.008 mg/kg in algae. The THg detection rates had increased after cooking, which were inshore fish (97%), other farmed fish (89%), shellfish (67%), cephalopods (100%), and crustaceans (88%). The mean THg concentrations were also increased.

Figure 1 summarizes the THg concentrations in the seven categories of raw and steam-cooked seafood. Samples were collected from fishing ports in the north, west, and south of Taiwan and from imported seafood. According to our results, among all pelagic fish samples, those collected from the fishing ports of southern Taiwan featured a higher THg concentration (Figure 1c); while imported inshore fish had higher levels of THg compared to other inshore fish samples (Figure 1e). Among the other farmed fish samples, those collected from fishing ports of northern Taiwan showed higher THg concentrations (Figure 1a). Taken together, the order of THg concentration from high to low is pelagic fish, inshore fish, and other farmed fish, while the THg concentrations in shellfish, cephalopods, crustaceans, and algae are much lower. The detected THg concentrations in the northern and eastern Taiwan were higher than that in other regions, it is possible related to the source of seafood in the area.

### 3.2. Human Health Risk Assessment

A MC simulation was employed to determine C, CR, and BW due to the sparse data. The exposure groups were classified according to different age groups as follows: 1–3, 4–6, 7–18, ≥19 years old, and women of childbearing age (19–49 years old). Figure 2 lists the risks of consuming the seven seafood categories for different population groups. According to the most conservative 100%PTWI results at the 95th percentile, the 1–3 and 4–6 age groups were at higher risk than the age groups of 7–18 and ≥19, as well as women of childbearing age (19–49 years old). However, the long tail suggested that a relatively high level of uncertainty was present. If the results at the 75th percentile were considered, all age groups were below 100%PTWI, meaning the risks were all acceptable.

Figure 3 shows the sum of the risks at the 50th and 75th percentiles in different seafood categories for the different population groups. Figure 3a shows the results of the summed total risk at the 50th percentile (the general situation) of the seven seafood categories, and all the risks for population groups were within 100%PTWI, suggesting that the risks were acceptable. Figure 3b shows the summed total risks at the 75th percentile, and under this conservative scenario, the risk levels for population groups of children 1–3 years and 4–6 years were both higher than 100%PTWI, suggesting that follow-up studies are needed, and extremely high consumers should pay attention to the MeHg hazard-related risks and moderately change their habits of consuming seafood.

### 3.3. Risk and Nutrition-Based Recommended Weekly Consumption Advisory

Chen et al. [39], Hsi et al. [5], and Mahaffey et al. [40] indicated that some pelagic and inshore fish feature higher levels of MeHg, but lower levels of EPA and DHA. Del Gobbo et al. [42] also suggested that some pelagic fish should be avoided or consumed less often, and that the recommended amount for children and childbearing women should be < 75 g/month. Therefore, we fixed the recommended intake of pelagic and inshore fish to determine the recommended weekly consumption advisory for other farmed fish. This study established fish consumption advisories for sensitive population groups, including 1–3-year-old children, 4–6-year-old children, and 19–49-year-old childbearing women. This study set up hypothetical scenarios for the different population groups where the consumption of pelagic fish and inshore fish were set to a fixed number of servings (e.g., in Scenario 3 for 1–3-year-old children, pelagic fish were not consumed and one serving (35 g per serving) of inshore fish was consumed; in Scenario 3 for 19–49-year-old childbearing women, two servings (70 g) of pelagic fish and seven servings of inshore fish were consumed. 

By removing the risks from pelagic fish and inshore fish based on the 100%PTWI at the 75th percentile, the remaining risk could be used to make inferences, and weekly consumption advisories could be made about the maximum recommended consumption of other farmed fish for the different population groups (Table 5). If, in Scenario 3, the group aged 1–3 years consumed no pelagic fish and one serving (35 g) of inshore fish, the maximum recommended consumption advisory for other farmed fish would be 197.3 g. If the group aged 4–6 years in Scenario 2 consumed no pelagic fish and one serving (35 g) of inshore fish, the maximum recommended consumption advisory for other farmed fish would be 653.3 g. If the group of 19−49-year-old women in Scenario 3 consumed two servings of pelagic fish and seven servings of inshore fish, the maximum recommended consumption advisory for other farmed fish would be 189.4 g.

According to the recommendation of the Food and Agriculture Organization (FAO)/WHO [43], the recommended daily intake (RDI) for EPA and DHA in 1–3-year-olds is 150 mg/day, for 4–6-year-olds the RDI is 200 mg/day, and in 19–49-year-old women the RDI is 250 mg/day; the mean concentration of EPA and DHA of pelagic fish, inshore fish, and other farmed fish, which were collated from studies of Afonso et al. [44], Cantoral et al. [45], Chen et al. [39], Del Gobbo et al. [42], Hsi et al. [5], Mahaffey et al. [40], Laird et al. [8], Rincón-Cervera et al. [46], were 3.892, 2.989, and 7.922 (mg/g). Collated mercury concentrations and EPA+DHA concentrations in different fishes are presented in Appendix A. This study assumed that 100% of the EPA and DHA consumed originate in seafood and calculated the minimum recommended consumption of other farmed fish that equaled the RDI of EPA and DHA. After removing the EPA and DHA intake from pelagic fish and inshore fish from the RDI, then dividing by the EPA and DHA concentration found in other farmed fish, we can obtain the minimum recommended consumption of other farmed fish (Table 5). If the group aged 1–3 years in Scenario 3 consumed no servings of pelagic fish and one serving of inshore fish, the minimum recommended consumption advisory for other farmed fish would be 119.3 g. If the group aged 4–6 years in scenario 2 consumed no servings of pelagic fish and one serving of inshore fish, the minimum recommended consumption advisory for other farmed fish would be 163.5 g. If the group of women aged 19–49 years in Scenario 3 consumed two servings (70 g) of pelagic fish and seven servings (210 g) of inshore fish, the minimum recommended consumption advisory for other farmed fish would be 94.1 g. The German Federal Institute for Risk Assessment (BfR) has established 1500 mg/day as the recommended upper intake level for omega-3 polyunsaturated fatty acids [47], and the recommendation of the US Food and Drug Administration (USFDA) is to not exceed an intake of 3000 mg/day omega-3 fatty acids (EPA and DHA) [48]. Appendix B shows the daily intake of DHA and EPA based on the maximum consumption advisory of fish intake (Table 5) multiplied by the mean concentration of EPA and DHA, which exceeded the RDI suggested by the FAO/WHO [43], but did not exceed the RDI suggested by the BfR [47] or the USFDA [48].

### 3.4. Comparison of Raw and Cooked Food

Overall, Hg concentrations were higher in cooked samples than in raw samples, showing an increase of 50% to 80%. We speculate that the cooking process reduced the weight of the seafood, resulting in an enrichment of THg and MeHg. Among the literature concerning the heavy metal level changes before and after cooking seafood, one study showed that the concentration of arsenic (As) in shellfish, crustaceans, and cephalopods changed after cooking [49]; and another indicated an increase in As, cadmium (Cd), and lead (Pb) levels in sardines, cod, and tuna after grilling, however, the changes were not statistically significant [50]. Cooked seafood may undergo accelerated protein degradation and water loss due to the effects the process of heat treatment has on organic material, leading to elevated heavy metal levels [51]. In addition, Kalogeropoulos et al. found that the increase of heavy metal levels in smaller fish was more obvious due to a greater loss of water during cooking [52]. In contrast, Diaconescu et al. suggested that decreased levels of heavy metals [53], including chromium (Cr), nickel (Ni), and Pb, after baking and microwave cooking were probably the result of either a reduced binding of metal to the proteins in the tissues after the cooking process (Howarth and Sprague 1978), or the formation of free salt compounds after the heavy metals were partially removed along with the water loss [54]. Ersoy (2011) found that Cd and cobalt (Co) were not detected in either raw or cooked catfish, but the concentrations of As, Cr, and Ni significantly increased after cooking [54]. Taken together, the above results suggest that, apart from the differences between species, fish size, and cooking methods, water loss could be a major cause of THg level changes between raw and cooked seafood. In addition, THg cannot be reduced or completely removed from seafood by cooking, meaning that humans may still take in a certain level of THg, even from the consumption of cooked seafood. In the cases where the consumption risk of THg and MeHg need to be investigated in follow-up studies, one should consider the concentration changes in THg and MeHg in seafood before and after cooking, as well as the fact that different cooking methods may also cause differences in THg and MeHg levels. 

### 3.5. Correlation between Seafood Weight and THg Concentration

Figure 4 shows the correlation between THg concentration and weight of inshore fish, other farmed fish, shellfish, cephalopods, and crustaceans. Most values are expected in the inter quartile range (IQR). Values outside the boundaries of 3(IQR) are termed “extreme outliers”. In this figure, if any observation lies outside the range Q1 − (3 IQR) and Q3 + (3 IQR), they are defined as problematic outliers and removed from calculation.

According to the results, there was no significant correlation between the weight of an aquatic product and THg concentration. This could have been caused by the differences in aquatic food species and sampling locations. However, the correlation between the weight of other farmed fish and THg concentration was higher than for other seafood, and the changes in THg concentrations with the weight of other farmed fish and shellfish were more obvious, but there was no significant correlation. There was a negative correlation between the weight of shellfish and crustaceans and THg concentration, but there was no significant correlation.

## 4. Conclusions

This study provides a database for THg and MeHg concentrations in raw and cooked seafood, which includes pelagic fish, inshore fish, other farmed fish, crustaceans, shellfish, cephalopods, and algae. Pelagic fish have the highest average THg concentration, followed by inshore fish, other farmed fish, crustaceans, shellfish, cephalopods, and algae. The THg concentrations in cooked seafood were always higher than those in raw seafood. According to the results of the cumulative risk at the 50th percentile, the risk for all age groups was below 100%PTWI and thus was an acceptable risk. However, the risks for children 1–3 years and 4–6 years were above 100%PTWI at the 75th percentile, calling for further investigation. This study also advised recommended weekly consumption rates of seafood for the population groups of 1–3-year-olds, 4–6-year-olds, and 19–49-year-old women based on different hypothetical scenarios.

## Figures and Tables

**Figure 1 ijerph-18-12227-f001:**
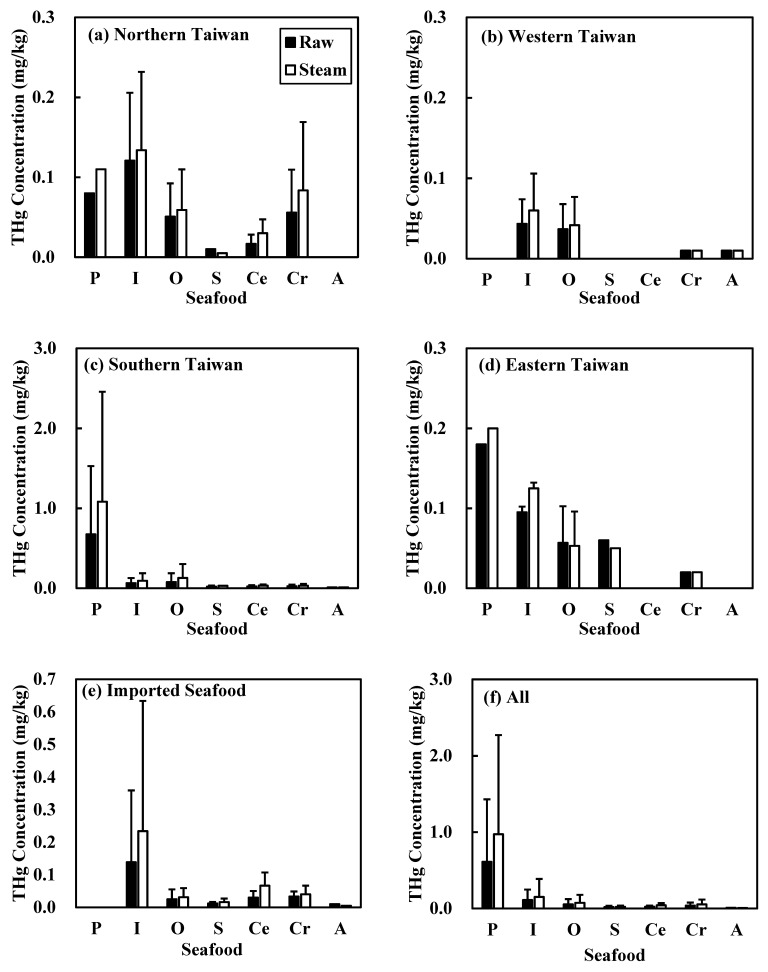
Concentrations of total mercury (THg) in seafood from different sources in (**a**) northern Taiwan, (**b**) western Taiwan, (**c**) southern Taiwan, (**d**) eastern Taiwan, (**e**) imported seafood, and (**f**) all purchased seafood. P: Pelagic fish; I: Inshore fish; O: Other farmed fish; S: Shellfish; Ce: Cephalopods; Cr: Crustaceans; and A: Algae.

**Figure 2 ijerph-18-12227-f002:**
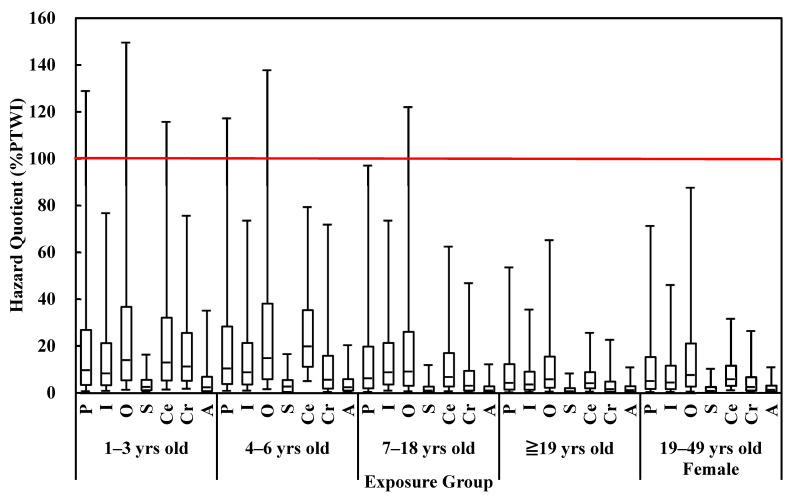
Hazard quotient for the seven categories of seafood among different age groups. P: Pelagic fish; I: Inshore fish; O: Other farmed fish; S: Shellfish; Ce: Cephalopods; Cr: Crustaceans; and A: Algae.

**Figure 3 ijerph-18-12227-f003:**
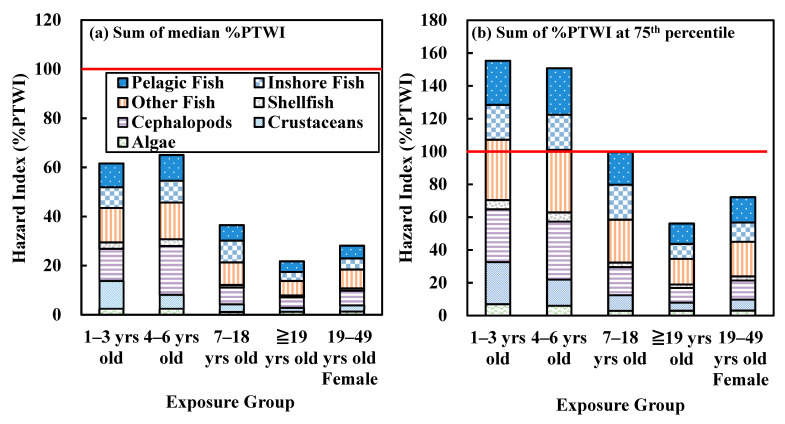
Hazard index of dietary exposure to MeHg in seafood for different age groups. (**a**) Sum of all median 100%PTWI, (**b**) Sum of all 100%PTWI at 75th percentile.

**Figure 4 ijerph-18-12227-f004:**
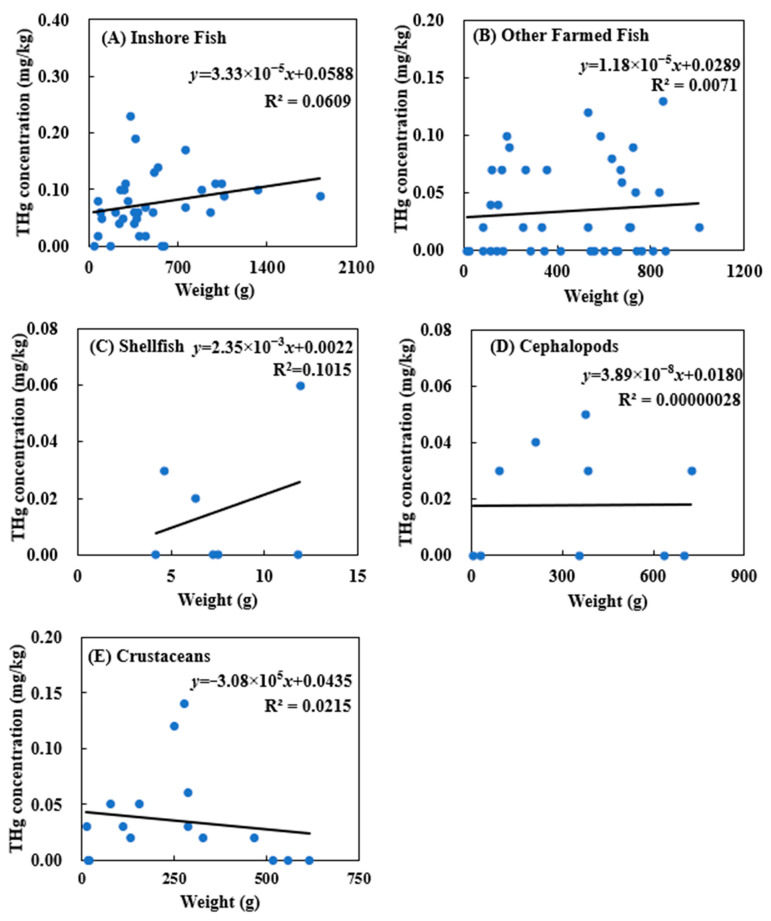
Correlation between total mercury concentration and weight of (**A**) inshore fish, (**B**) other farmed fish, (**C**) shellfish, (**D**) cephalopods, and (**E**) crustaceans.

**Table 1 ijerph-18-12227-t001:** Classification of the analyzed seafood samples.

Seafood Category	Common Name (Sample Size)
Pelagic Fish (*n* = 17)	Bastard Albacore (1), Bigeye Tunny (1), Blue Marlin (2), Marlin (1), Pacific Sailfish (1), Pointed Nose Shark (1), Requiem Shark (1), Snake Mackerel (2), Swordfish (4), Tribon Blou (2), and Yellow-Fin Tuna (1).
Inshore Fish (*n* = 35)	Anglerfish (2), Barracuda (3), Bullet Mackerel (1), Catfish (2), Cod (1), Cutlassfish (1), Eel (1), Freshwater Eel (2), Halibut (3), Pike Congers (2), Pompano (1), Red Seabream (2), Righteye Flounder (2), Scat (2), Silverfish (3), Skipjack (2), Stingray (3), Sturgeon (1), and Yellow Seabream (1).
Other Farmed Fish (*n* = 47)	Barramundi (3), Blue Mackerel (3), Butterfish (2), Cobia (1), Common Dolphinfish (1), Giant Grouper (2), Golden Threadfin Bream (2), Greater Amberjack (1), Japanese Butterfish (3), Japanese Horse Mackerel (1), Lizardfish (1), Milkfish (4), Moonfish (1), Oceanic Anchovy (1), Orange-Spotted Grouper (4), Round Scad (1), Salmon Trout (1), Saury (2), Sea Cucumber (1), Shishamo (2), Shrimp Scad (2), Silver-Stripe Round Herring (1), Sweet Fish (1), Tilapia (4), and Tilefish (1).
Shellfish (*n* = 9)	Babylonia (3), Clam (1), Freshwater Clam (1), Locos (1), Mussel (1), Oyster (1), and Variegate Venus (1).
Cephalopods (*n* = 10)	Argentine Shortfin Squid (2), Cuttlefish (1), Jumbo Flying Squid (1), Loligo Squid (1), Octopus (2), and Squid (3).
Crustaceans (*n* = 16)	Big Head Shrimp (3), Blue Crab (1), Crayfish (1), Giant River Prawn (1), Grass Shrimp (1), Lobster (2), Pelagic Crab (3), Sakura Shrimp (2), Serrated Crab (1), and Whiteleg Shrimp (1).
Algae (*n* = 6)	Eucheuma (1), Gracilaria Seaweed (1), Purple Laver (2), and Sea Tangle (2).

**Table 2 ijerph-18-12227-t002:** Consumption rates for the seven seafood categories among different exposure populations.

Seafood Category	Consumption Rates (g/Week) for Various Age Groups				
	1–3 Years Old	4–6 Years Old	7–18 Years Old	≥19 Years Old	19–49-Year-Old Women
Pelagic Fish	LN(9.24, 16.73) ^a^	LN(14.28, 15.68)	LN(19.32, 16.10)	LN(19.95, 16.87)	LN(20.86, 18.34)
Inshore Fish	LN(31.29, 16.73)	LN(48.30, 15.68)	LN(65.66, 16.10)	LN(67.55, 16.87)	LN(70.84, 18.34)
Other Farmed Fish	LN(123.90, 16.73)	LN(191.24, 15.68)	LN(259.70, 16.10)	LN(267.40, 16.87)	LN(280.42, 18.34)
Shellfish	LN(67.13, 16.52)	LN(102.06, 16.03)	LN(83.16, 21.77)	LN(107.45, 22.47)	LN(104.86, 24.36)
Cephalopods	LN(167.09, 21.14)	LN(371.00, 11.13)	LN(294.49, 16.24)	LN(277.76, 15.82)	LN(322.49, 14.14)
Crustaceans	LN(149.45, 13.09)	LN(102.83, 24.29)	LN(136.43, 20.86)	LN(107.45, 24.57)	LN(139.23, 19.74)
Algae	LN(64.05, 30.80)	LN(94.57, 22.12)	LN(90.30, 20.65)	LN(155.82, 22.75)	LN(147.00, 21.35)

^a^ LN(gm, gsd) represents lognormal distribution with geometric mean (gm) and geometric standard deviation (gsd).

**Table 3 ijerph-18-12227-t003:** Percentages of methylmercury to total mercury (% MeHg/THg) in different seafood species adopted from a previous study.

Species	Scientific Name	% MeHg/THg	Reference
Anglerfish	Anglerfish *Lophiomus setigerus*	92.5	[31]
Cephalopods	Argentine Shortfin Squid *Illex argentinus*	81.3	[32]
	Jumbo Flying Squid *Dosidicus gigas*		
	Cuttlefish *Sepia Esculenta*	72.8	[32]
	Octopus *Octopus vulgare*	92	[37]
Cod	Cod *Coelorinchus kamoharai*	97.4	[31]
Crustaceans	Big Head Shrimp *Macrobrachium rosenbergii*	93	[33]
	Grass Shrimp *Penaeus monodon*		
	Whiteleg Shrimp *Sergia lucens*		
Eel	Freshwater Eel *Anguilla luzonensis*	100	[36]
	Eel *Muraenesox cinereus*	83.6	[32]
Flatfish	Sole *Cynoglossus arel*	77.3	[32]
Hairtail	Cutlassfish *Trichiurus nanhaiensis*	99	[38]
	Silverfish *Trichiurus lepturus*		
Herring	Silver-Stripe Round Herring *Spratelloides gracilis*	100	[38]
Lizardfish	Lizardfish *Saurida tumbil*	100	[30]
Marlin	Blue Marlin *Makaira nigricans*	84	[34]
	Pacific Sailfish *Istiophorus platypterus*		
	Swordfish *Xiphias gladius*	99	[36]
Oilfish	Snake Mackerel *Gempylus serpens*	92	[36]
Pomfret	Butterfish *Pampus echinogaster*	75.4	[32]
Pompano	Pompano *Parastromateus niger*	94	[38]
Salmon	Salmon Trout *Oncorhynchus mykiss*	93	[37]
Saury	Saury *Saurida undosquamis*	75	[36]
Shark	Requiem Shark *Carcharhinus melanopterus*	73	[34]
	Tribon Blou *Prionace glauca*		
Shellfish	Oyster *Crassostrea gigas*	82	[33]
	Mussel *Perna viridis*	35	[36]
Snapper	Black Sea Bream *Acanthopagrus schlegelii*	100	[38]
	Red Seabream *Pagrus major*		
	Yellow Seabream *Dentex hypselosomus*		
Tilapia	Tilapia *Oreochromis niloticus*	95	[35]
Tuna	Bastard Albacore *Thunnus alalunga*	93	[37]
	Bigeye Tunny *Thunnus obesus*		
	Yellow-Fin Tuna *Thunnus albacares*		

**Table 4 ijerph-18-12227-t004:** Total mercury (THg) concentrations in raw (uncooked) and cooked seafood.

Seafood Categories	Number of Species	Uncooked (Raw)				Cooked (Steam)			
		Number of Samples	≥LOQ	THg Concentration (mg/kg)		Number of Samples	≥LOQ	THg Concentration (mg/kg)	
				Mean ± SD	Range			Mean ± SD	Range
Pelagic Fish	11	17	17	0.61 ± 0.82	0.03–3.16	17	17	0.97 ± 1.30	0.05–4.59
Inshore Fish	19	35	32	0.11 ± 0.14	ND ^a^–0.78	35	34	0.15 ± 0.24	ND ^b^–1.39
Other Farmed Fish	25	47	29	0.05 ± 0.07	ND ^a^–0.35	47	42	0.07 ± 0.11	ND ^b^–0.47
Shellfish	7	9	3	0.02 ± 0.02	ND ^a^–0.06	9	6	0.02 ± 0.02	ND ^b^–0.05
Cephalopods	6	10	5	0.02 ± 0.01	ND ^a^–0.05	10	10	0.04 ± 0.03	0.02–0.11
Crustaceans	10	16	11	0.04 ± 0.04	ND ^a^–0.14	16	14	0.05 ± 0.06	ND ^b^–0.23
Algae	4	6	1	0.01 ± 0.00	ND ^a^	6	1	0.01 ± 0.00	ND ^b^–0.01

^a^ ND: 0.02 mg/kg for uncooked (raw) seafood ^b^ ND: 0.01 mg/kg for cooked (steamed) seafood.

**Table 5 ijerph-18-12227-t005:** Consumption advisories for fish intake (g/week).

Seafood Category	Weekly Consumption Advisory (g/Week)							
	1–3 Years Old			4–6 Years Old		19–49-Year-Old Women		
	Scenario 1	Scenario 2	Scenario 3	Scenario 1	Scenario 2	Scenario 1	Scenario 2	Scenario 3
Pelagic Fish	0.0	0.0	0.0	0.0	0.0	70.0	70.0	70.0
Inshore Fish	8.8	17.5	35.0	17.5	35.0	175.0	210.0	245.0
Other Farmed Fish	129.2–453.0	125.9–366.1	119.3–197.3	170.1–788.5	163.5–653.3	120.5–855.5	107.3–527.8	94.1–189.4

## Data Availability

Not applicable.

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
