# Peer review of "Dietary Exposure of the Taiwan Population to Mercury Content in Various Seafood Assessed by a Total Diet Study"

_ijerph, 2021, doi:10.3390/ijerph182212227_

Round 1
Reviewer 1 Report
Observations
- Indicate what PTWI means (line 148).
- How can the HQ values be expressed in% PTWI, if PTWI is part of equation 1?
- Explain why LOD / 2 for undetectable values was considered(line 202).
- Carry out a discussion of values presented in table 4, not only mention the values again (lines 216 to 229).
- Use different shading for the columns of the graphs to differentiate between raw and cooked samples (figure1).
- Conduct further discussion of Figures 1, 2 and 3
- Use different shading in the graphs to differentiate the groups (figure ).
- Table 5 does not distinguish the ranges for the values of "Other Farmed Fish"
- Indicate what the risk is according to the actual consumption in the different groups of people.
- There is a need for discussion on consumption according to the different regions.
Author Response
We appreciate reviewer for providing these valuable comments. The comments could greatly improve the quality of this manuscript. The followings are the detailed responses to each individual comment.
1. Indicate what PTWI means (line 148); How can the HQ values be expressed in% PTWI, if PTWI is part of equation 1?
Response: We have revised the paragraph to indicate PTWI meaning clearly. (P4, L148-153), and explained the calculated of calcium intake (P4, Equation 1).
2. Explain why LOD / 2 for undetectable values was considered(line 202).
Response: To avoid the situation which the analytical results were ND, but actually existed just were lower than the LOD. In order to conservatively estimate the concentration, we refer to the assumptions applied by GEMS/Food ERUO, if the proportion of analytical results which were non-detectable (ND) was less than 60%, replacing the ND results with LOD/2. (P6, L201-205).
3. Carry out a discussion of values presented in table 4, not only mention the values again (lines 216 to 229).
Response: Thank you for kind comment, we have added a discussion of the values in Table 4 (P7, L227-232; P7, 239-241).
4. Use different shading for the columns of the graphs to differentiate between raw and cooked samples (figure1). Conduct further discussion of Figures 1, 2 and 3.
Response: Thank you for kind comment, we have revised the columns into black and white to differentiate between raw and cooked samples (P9, Figure 1). Further discussion of Figures 1 revised in (P8, 251-253)
5. Table 5 does not distinguish the ranges for the values of "Other Farmed Fish"
Response: Thank you for kind comment, we have revised the range for the values of Table 5 (P12, Table 5).
6. Indicate what the risk is according to the actual consumption in the different groups of people.
Response: The endpoints of MeHg have been assessed of neurodevelopmental evaluated by WHO (ref. 29), and gave a value of 1.6 µg/kg bw for PTWI. Therefore, there is no special description in the text.
7. There is a need for discussion on consumption according to the different regions.
Response: This study aims to assess the MeHg intake risk of all people in Taiwan, so there is no special discussion on consumption according to the different regions of Taiwan.
Reviewer 2 Report
“Dietary Exposure of the Taiwan Population to Mercury Content in Various Seafoods Assessed by a Total Diet Study” is very interesting and well designed study. The topic is significant for public as well as for scientific community. The results are clearly presented and conclusions are sound and well explained.
I have few questions and suggestions, but I understand that you can not act upon them since the research is already finished. You have a large number of samples in your study which is praiseworthy, but it is still just one sample per animal species. Would you consider making composite samples in the future? In that way you can get more representative results with the same number of analyses.
Some fish which you have analyzed are more often consumed than others. Could you adjust your risk assessment, so it would give more weight to the results of the fish which is more often consumed and in larger quantities?
l143 Can you describe steam cooking procedure in more detail?
l144 How was the sample prepared (digested) prior to ICP-MS analysis?
l145 Can you describe ICP-MS measurements in more detail?
l146 Which CRM have you used?
l148 Please define PTWI
l164 (Table 2) Is it possible to add uncertainties to the values stated in the table?
l167 (Table 3) Can you explain what are the numbers in brackets?
l346 “In addition, THg cannot be reduced or completely removed from seafood by cooking, meaning that humans may still take in a certain level of THg even from the consumption of cooked seafood.” As I understood your research as well as the previous studies found the opposite trend from what you wrote. Why would you mention option that THg might be completely removed from seafood by cooking?
l357 “This could have been caused by the differences in aquatic food species and sampling locations.” Why would mentioned difference cause the lack of correlations?
l359 higher instead of high?
l358-360 Could you check again these lines. You don’t mention inshore fish, but you mention other farmed fish twice. Is it mistake or I didn’t understand something well?
Fig 4(a) The dot in the left upper corner is obviously outlier and if you have used Pearson’s correlation coefficient it has significantly affected your results. You should either exclude it or use some nonparametric coefficient instead (Kendall or Spearman). All graphs in the figure have the same scale (0-1 mg/kg). I understand your logic, but I think that this way of presenting is not useful, especially for graphs (c) to (e).
Why is pelagic fish omitted from the correlation analysis? I guess because you have bought only parts of the fish meat without knowing how big the fish was? Could you find average size of these species of fish and perform analysis based on that (just state it in the text).
Please keep in mind that bioconcetration and biomagnification are main factors which lead to elevated content of Hg in fish. This means that fish which is older and higher in the food chain will have higher Hg concentrations. Mentioned traits are often overlapped with size of the fish (older and predator fish tend to be more massive), but not necessarily in every case. Many other factors such as contact with sediments, metabolism etc. may also significantly affect Hg content in fish.
Author Response
We appreciate reviewer for providing these valuable comments. The comments could greatly improve the quality of this manuscript. The followings are the detailed responses to each individual comment.
1. You have a large number of samples in your study which is praiseworthy, but it is still just one sample per animal species. Would you consider making composite samples in the future? In that way you can get more representative results with the same number of analyses.
Response: This study is based on the Constructing a Diet Representative List. After purchasing aquatic products from various places in Taiwan, the samples are mixed and homogenized. That is, a sample is mixed with aquatic products from multiple regions, errors can be reduce. If we want to conduct follow-up studies in the future, we can Sampling towards aquatic product categories with higher calculated risks.
2. Some fish which you have analyzed are more often consumed than others. Could you adjust your risk assessment, so it would give more weight to the results of the fish which is more often consumed and in larger quantities?
Response: The more frequently consumed aquatic products also have higher consumption rate, so the calculated risk will also be higher.
3. L143 Can you describe steam cooking procedure in more detail?
Response: Steam-cooked samples were put in the pot and steam for 10 minutes, we have added the paragraph in (P4, L140).
4. L144 How was the sample prepared (digested) prior to ICP-MS analysis?; L145 Can you describe ICP-MS measurements in more detail? ; L146 Which CRM have you used?
Response: This research focuses on the risk assessment of MeHg for Taiwanese. Due to space limitations, we choose to focus on risk calculation.
5. L148 Please define PTWI
Response: We have revised the paragraph to indicate PTWI meaning clearly. (P4, L148-153), and explained the calculated of calcium intake (P4, Equation 1).
6. L164 (Table 2) Is it possible to add uncertainties to the values stated in the table?
Response: There is no uncertainty about the value in these references.
7. L167 (Table 3) Can you explain what are the numbers in brackets?
Response: The numbers in brackets are geometric mean and geometric standard deviation.
8. L346 “In addition, THg cannot be reduced or completely removed from seafood by cooking, meaning that humans may still take in a certain level of THg even from the consumption of cooked seafood.” As I understood your research as well as the previous studies found the opposite trend from what you wrote. Why would you mention option that THg might be completely removed from seafood by cooking?
Response: We have revised the word “removed” into “reduced” (P12, L341).
9. L357 “This could have been caused by the differences in aquatic food species and sampling locations.” Why would mentioned difference cause the lack of correlations?
Response: The collected samples may have different MeHg concentrations due to contamination in different areas.
10. L359 higher instead of high?
Response: Thank you for kind comment, we have revised the word into higher.
11. L358-360 Could you check again these lines. You don’t mention inshore fish, but you mention other farmed fish twice. Is it mistake or I didn’t understand something well?
Response: Weight of other farmed fish and THg concentra-tion was higher than other seafoods
12. Fig 4(a) The dot in the left upper corner is obviously outlier and if you have used Pearson’s correlation coefficient it has significantly affected your results. You should either exclude it or use some nonparametric coefficient instead (Kendall or Spearman). All graphs in the figure have the same scale (0-1 mg/kg). I understand your logic, but I think that this way of presenting is not useful, especially for graphs (c) to (e).
Response: Thank you for kind comment, we have revised the figure and the text, in this study, if any observation lies outside the range Q1 – (3 IQR) and Q3 +(3 IQR) it is defined as problematic outliers and remove from calculation. (P13, L369-372).
13. Why is pelagic fish omitted from the correlation analysis? I guess because you have bought only parts of the fish meat without knowing how big the fish was? Could you find average size of these species of fish and perform analysis based on that (just state it in the text).
Response: It is true that we cannot measure the complete weight of the large pelagic fishs we bought. If we want to measure the data, we can only wholesale the whole fish, but the price is extremely high.
14. Please keep in mind that bioconcetration and biomagnification are main factors which lead to elevated content of Hg in fish. This means that fish which is older and higher in the food chain will have higher Hg concentrations. Mentioned traits are often overlapped with size of the fish (older and predator fish tend to be more massive), but not necessarily in every case. Many other factors such as contact with sediments, metabolism etc. may also significantly affect Hg content in fish.
Response: Thank you for your reminder. In this research, we discuss the consumption of aquatic products in Taiwanese. As you said, many other factors such as contact with sediments. This is why we collect so many kinds of aquatic products.
Reviewer 3 Report
The authors measured Hg concentrations of seafood regularly consumed in Taiwan and estimated Hg exposure rate for different group of population. Based on the results, the authors further advised consumption rate of different seafood for different populations. The research strategy is OK. If this study were well conducted, from sampling, to laboratory measurements, to data analysis, to results presentation, this work could have become high-quality research. However, there are many flaws/deficiencies in this study, in almost every step (i.e., sample collection, Hg measurements, data analysis, manuscript writing). For example, the researchers targeted an ambitious sampling species of 82, but only collected 140 samples. It turned out most species only had one or two samples, so the representativeness is questionable. Quite a large fraction of samples with Hg concentrations lower than LOD greatly limited the value of the data. Failure in correlating seafood body weight with Hg concentrations likely resulted from the sampling plan, which had too many species yet too few samples. Writing itself is in low quality, both scientific wise and English wise. Many statements in Introduction were taken for granted and not really true. All these flaws and deficiencies made current format of manuscript almost valueless. I would suggestion this manuscript be rejected.
Author Response
We appreciate reviewer for providing these valuable comments. The comments could greatly improve the quality of this manuscript. The followings are the detailed responses to each individual comment.
The authors measured Hg concentrations of seafood regularly consumed in Taiwan and estimated Hg exposure rate for different group of population. Based on the results, the authors further advised consumption rate of different seafood for different populations. The research strategy is OK. If this study were well conducted, from sampling, to laboratory measurements, to data analysis, to results presentation, this work could have become high-quality research. However, there are many flaws/deficiencies in this study, in almost every step (i.e., sample collection, Hg measurements, data analysis, manuscript writing). For example, the researchers targeted an ambitious sampling species of 82, but only collected 140 samples. It turned out most species only had one or two samples, so the representativeness is questionable. Quite a large fraction of samples with Hg concentrations lower than LOD greatly limited the value of the data. Failure in correlating seafood body weight with Hg concentrations likely resulted from the sampling plan, which had too many species yet too few samples. Writing itself is in low quality, both scientific wise and English wise. Many statements in Introduction were taken for granted and not really true. All these flaws and deficiencies made current format of manuscript almost valueless. I would suggestion this manuscript be rejected.
Response: Thank you for comment, we have corrected the errors in the grammar. In the sampling part, we focused on using TDS to assess the risk of consuming different types of aquatic products, and the number of aquatic products of seafood categories were Pelagic Fish (17), Inshore Fish (35), Other Farmed Fish (47), Shellfish (9), Cephalopods (10), Crustaceans (16), and Algae (6) are all more than one sample. If follow-up research needs to be more, then it can be focused on a single species.
Reviewer 4 Report
Review comments on the manuscript by Lin et al
Dietary exposure of the Taiwan population to mercury content in various seafoods assessed by a total diet study
The aim of the study detailed in this manuscript was to assess the health risks of exposure to methylmercury in contaminated seafood on the diet of people in Taiwan. In the study, methylmercury concentrations were analysed for seven types of seafood from a variety of locations across Taiwan. They also assessed the effects of cooking on methylmercury concentrations. A range on consumption scenarios for different at-risk age groups in the population were also assessed. Based on the results of the seafood analysis for methylmercury and the scenario testing, they then provided weekly seafood consumption advisories for the various at-risk age groups.
Overall, the data presented achieves the study aims and provides an original contribution to our knowledge of the methyl mercury concentrations in seafood from Taiwan and the variability between different seafood types and regions. The analytical techniques and QA/QC procedures, sampling design and statistical analysis are appropriate to address the aims of the study and the tables and figures are relevant to presenting the results of the study. As such, I consider that a revised manuscript, incorporating the marked comments on the PDF and addressing the comments detailed below, would be suitable for publication.
The authors should replace the text for this paragraph in the manuscript with the test below to improve the sentence structure.
3.3. Risk and Nutrition-Based Recommended Weekly Consumption Advisory
Chen et al. [30], Hsi et al. [5], and Mahaffey et al. [31] indicated that some pelagic and inshore fish feature higher levels of MeHg, but lower levels of EPA and DHA. Del Gobbo et al. [42] also suggested that some pelagic fish should be avoided or consumed less often, and that the recommended amount for children and childbearing women should be <75 g/month. Therefore, we fixed the recommended intake of pelagic and inshore fish to determine the recommended weekly consumption advisory for other farmed fish. This study established fish consumption advisories for sensitive population groups, including 1–3 years old children, 4–6 years old children, and 19–49 years old childbearing women. This study set up hypothetical scenarios in different population groups where the consumption of pelagic fish and inshore fish were set fixed servings (e.g. in Scenario 3 for 1–3 years old children, pelagic fish were not consumed and one serving (35 g per serving) of inshore fish was consumed; in Scenario 3 for 19–49 years old childbearing women, two servings (70 g) of pelagic fish and seven servings of inshore fish were consumed.
The authors should provide a reference for the data in Table 2 as it is not currently obvious in the text.
Given that the author’s first language is not English, I have marked as track changes on the manuscript in the attached PDF, numerous suggested spelling and grammatical changes to improve the quality and flow of the manuscript. The authors should review these changes and have the manuscript reviewed before resubmission.

Author Response
We appreciate reviewer for providing these valuable comments. The comments could greatly improve the quality of this manuscript. The followings are the detailed responses to each individual comment.
1. The authors should replace the text for this paragraph in the manuscript with the test below to improve the sentence structure.
Response: Thank you for kind comment. We have revised the text to make clearly (P4, L134-143).
2. The authors should provide a reference for the data in Table 2 as it is not currently obvious in the text.
Response: We have added references in the right column, and the text description added in (P4, L155-157).
3. Given that the author’s first language is not English, I have marked as track changes on the manuscript in the attached PDF, numerous suggested spelling and grammatical changes to improve the quality and flow of the manuscript. The authors should review these changes and have the manuscript reviewed before resubmission.
Response: Thank you for kind suggestion and comment. We have revised the title to make clearly.
Round 2
Reviewer 1 Report
Requested observations were made correctlyAuthor Response
Authors’ Response:
Thank you very much for your valuable comments and suggestions on our manuscript.
Reviewer 3 Report
The minimal edits barely improved the quality of this manuscript. In the reply the authors explained that more than one samples were collected for each category of seafood. However, trophic levels, determined by the biota species instead of the categorization used, largely controls Hg species concentrations. Unfortunately, the extremely small sample size (one or two for most biota species) led to the questionable representativeness. The deficiency in sampling likely resulted in failures in correlating seafood body weight with Hg concentrations, and the large standard deviation for Hg concentration in different seafood. Also, other major flaws and deficiencies pointed out by the original comments remained unaddressed. This manuscript did not reach the publishing standards for this journal, thus the reviewer recommend rejection of this manuscript.
Author Response
Authors’ Response:
Thank you very much for your valuable comments and suggestions on our manuscript. Based on the spirit of total diet study (TDS) method, the data of TDS differ from other chemical surveillance programs because they focus on chemical compounds in the diet rather than in individual foods. To assess all the seafood that Taiwanese might be intake, we have to ensure the seafood groups was most people eat. These representative seafood items were selected to be representative of the diet from the population in the region of Taiwan. Three main criteria were considered for selecting the seafood in the study: (1) the aquatic animal category in the Sanitation Standards for Aquatic Animals in Taiwan had to be a major item. (2) the Sanitation Standards for Algae Foods in Taiwan, the algae had to be a major item. (3) based on domestic production and imports minus exports in the Fisheries Industry Statistics in Taiwan. Products were ranked from high to low by domestic sales volume (tons). Therefore, according to the Fisheries Industry Statistics in Taiwan, we purchased non-imported seafood in the cities and counties with the three highest rankings for sales volume, among which were the fish species that people consumed frequently. We consider the total dietary risks were representative of calculating the concentration of mixed samples with the 7 category and the consumption rate of the 7 categories of aquatic products.